# Attention to detail: A conditional multi-head transformer for traffic sign recognition

**Isra Naz**[1,2], **Jamal Hussain Shah**[1], **Ali Tahir**[3], **Mahatma Reddy Marri**[4], **Rabia Saleem**[5], **Mutaz Elradi S. Saeed**[6]*

1 Department of Computer Science, COMSATS University Islamabad, Wah Campus, Islamabad, Pakistan, 2 Department of Computer Science, University of Wah, Wah Cantt, Pakistan, 3 Department of Computer Science, College of Engineering and Computer Science, Jazan University, Jazan, Kingdom of Saudi Arabia, 4 Independent Researcher, Texas,United States of America, 5 Department of Information Technology, Government College University Faisalabad, Gurunanakpura, Faisalabad, Pakistan, 6 Department of Computer Science, Nile Valley University, Atbara, Sudan

* mutelr@nilevalley.edu.sd

## Abstract

The challenge of traffic sign detection and recognition for driving vehicles has become more critical with recent advances in autonomous and assisted driving technologies. Although object recognition problems, particularly traffic sign recognition, have been extensively studied, most Vision Transformer (ViT) models still rely on static attention mechanisms with fixed projection matrices (Q, K, and V). Using this mechanism limits the ViTs to handle real-world problems such as object detection and traffic sign recognition, etc. Problems, such as partially or fully obscured signs, changes in illumination, and weather conditions, result in subpar feature extraction, which compounds the misclassification problem. To overcome this challenge, a Conditional Visual Transformer (CViT) is proposed in this research, which dynamically adapts feature aggregation, Q, K, and V projections, as well as attention-based mechanisms, based on the input sign type. Its main component consists of a controlled failure deep learning model using a CViT that targets specific types of traffic signs through varying feature extraction and attention adjustments, resulting in high classification performance and minimizing misclassifications. Furthermore, an adaptive gating technique is employed that optimally adjusts the projection matrix across different traffic signs. The proposed CViT achieved an overall accuracy of 99.87%, with a Micro Precision of 99.07%, a Macro Recall of 94.3%, and a Macro F1 Score of 99.07%, respectively. These results demonstrate the potential of CViT to improve both the efficiency and reliability of traffic sign recognition in autonomous driving applications.

**Data availability statement:** The dataset supporting this study is publicly available in the Figshare repository at https://www.doi.org/10.6084/m9.figshare.30373087. The source code is available at: https://github.com/IsraNaz786/Conditional-Vision-Transformer. In addition, source code is also uploaded in supporting information in .zip file.

**Funding:** The author(s) received no specific funding for this work.

**Competing interests:** The authors have declared that no competing interests exist.

**Abbreviations:** CViT, Conditional Visual Transformer; ViT, Vision Transformer; RGB, Red Green Blue (color channels); MHA, Multi-Head Attention; FFN, Feed-Forward Network; ReLU/ σ, Rectified Linear Unit/ Activation Function; FC, Fully Connected (Layer)

## 1 Introduction

The rapid development of autonomous vehicles has underscored the critical importance of computer vision systems, especially for accurate traffic sign recognition. Traffic Sign Recognition (TSR) enables fully automated driving systems to understand and respond appropriately to road signs and regulations [1]. Unfortunately, deep learning and even convolutional neural networks (CNNs) have not been able to fully overcome the problem of misclassification [2]. In the case of TSR, misclassification can have a devastating impact, from collisions and legal infractions to undermining the trust of the public in autonomous technology [3]. While current deep learning models excel in achieving high accuracy rates, their reliance on traditional performance metrics, such as confusion matrices, often overlooks the need to minimize misclassification. All classifiers, including CNNs, rely on constantly providing an output regardless of the threshold of their prediction confidence [4]. This non-information-preserving approach works for many applications; however, in safety-critical scenarios such as autonomous driving, an uncertain or misplaced label poses unacceptable risks.

Although CNNs have been the most dominant TSR applications for the last decade [5], they struggle to capture global dependencies in more complex and intricate visual environments. These limitations are further amplified in real-world conditions such as occlusions, illumination changes, or motion blur [6]. Moreover, decision-making and motion planning frameworks increasingly rely on accurate perception modules, where errors in traffic sign recognition can propagate into unsafe maneuvers [7].

Transformer-based architectures, originally developed for natural language processing, have recently been adapted for vision tasks to address these challenges [8]. These models employ self-attention methods designed to capture long-range dependencies, making them more effective at visual understanding [9]. Nonetheless, Visual Transformers (ViTs) overlook context capturing, meaning that all input tokens are treated homogeneously, which is not the best solution, especially in noisy or cluttered backgrounds. The risk of misclassification remains a significant concern, particularly in safety-critical scenarios, such as autonomous vehicles. Incorrect recognition of a traffic sign can lead to erroneous decision-making, potentially endangering passengers and pedestrians. To address these limitations, we propose a Conditional Vision Transformer (CViT) with an integrated fail-control mechanism. Unlike conventional classifiers that always produce an output, the fail-control mechanism withholds predictions when confidence is low, thereby preventing harmful errors. This capability enables the model to distinguish between confidently classifiable samples and those that should be acknowledged for further verification or human oversight.

## 2 Research contributions

The contributions of this study are as follows:

- Unlike standard ViTs that use a fixed cls_token, the proposed CViT dynamically chooses between the cls_token and Global Average Pooling (GAP) to ensure optimal feature representation across diverse traffic sign categories.

- Instead of fixed projection matrices, CViT employs a gating mechanism that learns adaptive Q, K, and V representations conditioned on the input sign type, thereby enhancing classification robustness.

- CViT integrates an adaptive attention mechanism that adjusts attention weights based on traffic sign characteristics (e.g., shape, color, or partial occlusion). This specialization ensures that each sign type receives tailored feature processing, reducing misclassifications and improving recognition accuracy.

The remainder of this paper is organized as follows: Section 3 provides background information on ML classifiers and their limitations in critical systems. The proposed software architecture for the proposed systems is presented in Section 4. The experimental setup and results are detailed in Section 5, while Section 6 discussesfuture work and concludes the paper.

## 3  Related work

TSR is a critical component of autonomous driving systems, enabling vehicles to interpret and respond to road regulations effectively. Recent advances in deep learning, particularly CNNs, have significantly improved the accuracy and robustness of TSR systems. [10] illustrated how modern CNN architectures such as VGG16 and EfficientNet have also been at the forefront in all other benchmark datasets, including the German Traffic Sign Recognition Benchmark (GTSRB). However, as with most innovations, there are a few glaring issues. In this case, TSR systems are sensitive to drastic changes and misclassifications, especially from everyday circumstances of illumination, occlusions, and even adversarial attacks that degrade performance [11,12]. The integration of TSR systems into autonomous vehicles has also been extensively studied. As noted in [13], real-time processing and risk profile of autonomous vehicles are of concern for TSR systems, stressing the importance of risk management throughout the AI lifecycle regarding the technologies of the use case. Although adversarial training and augmentation [14] have enhanced robustness, most CNN-based approaches still treat misclassification as secondary, focusing primarily on accuracy rather than reliability, an omission that is particularly dangerous in safety-critical contexts. Traditional evaluation metrics, such as accuracy, precision, and recall, focus on correctly classified instances but fail to adequately address the impact of misclassification [15,16]. This limitation is particularly problematic in safety-critical applications, such as autonomous driving, where even a single misclassification can lead to catastrophic outcomes. Recent work by [17] introduced tailored data augmentation strategies to address class imbalance and instance scarcity. While these methods improve robustness by enriching training data, they do not alter the decision process itself. [18] employed convolutional autoencoders as a dual-purpose defense, detecting adversarial perturbations and reconstructing corrupted inputs. While effective against attacks like FGSM and PGD, such defenses focus on input sanitization rather than output reliability. Similarly, localization-oriented models like RID-LIO have achieved robust LiDAR-based SLAM performance in degraded environments, highlighting the need for dependable perception in all conditions [19]."

**Takeaway 1:** CViT introduces dynamic attention and conditional projections (Q, K, V) to overcome the limitations of traditional Vision Transformers in handling real-world traffic sign recognition challenges

Recent research has shifted toward transformer-based models, specifically vision transformers (ViTs) [20], to overcome the limitations of traffic sign detection and recognition models, which leverage self-attention mechanisms to capture long-range dependencies and global contextual information. ViTs are emphasized here because their patch-based representation and self-attention enable them to capture global context that CNNs typically miss. This makes them particularly promising for TSR, where signs may appear in cluttered, partially occluded, or dynamically changing environments. Unlike CNNs, ViTs treat images as patches and process them similarly to natural language sequences, enabling a more holistic understanding of visual patterns. [21] proposed a Tokens-to-Token ViT architecture, which refines the patch embedding process and improves ImageNet training stability from scratch. Yin et al. [22] introduced A-ViT, an adaptive token sampling strategy to reduce computational overhead while maintaining accuracy. These models have paved the way for the application of ViTs in dense classification tasks such as TSR. Several researchers have explored ViT-based models to address real-world challenges in the domain of traffic sign recognition. In addition, [23] presented a local ViT tailored for

TSR, achieving superior results compared to traditional CNN models, particularly under conditions with high intraclass similarity and environmental noise. Similarly, [24] proposed an attention-guided CAM method that enhanced model explainability by utilizing self-attention weights to more precisely localize images. [25] proposed TSD-YOLO, a robust detection model that integrates ViTs within a YOLO framework to improve performance under adverse weather conditions such as fog, rain, and snow. Hybrid models that combine the strengths of CNNs and transformers have also gained attention [26]. Ghouse et al. [27] compared standard CNNs with transfer learning-based ViTs and concluded that ViTs offer better generalization. However, ViTs are not without drawbacks. Their reliance on large datasets and extensive pretraining limits scalability in real-world TSR deployments. Moreover, treating all patches equally overlooks context-dependent importance, which can reduce interpretability in safety-critical settings. To address this issue, researchers have explored knowledge distillation, data augmentation, and token pruning techniques. Beyond vision-based methods, multimodal frameworks such as YCANet have demonstrated the effectiveness of combining camera and LiDAR data for robust target detection in complex traffic scenes [28].

[29] explored cross-domain few-shot in-context learning with multimodal large language models (MLLMs) to reduce reliance on extensive labeled data. This improves generalization across different countries' traffic sign datasets but does not address the reliability of outputs under uncertainty. [30] proposed E-MobileViT, a lightweight ViT variant that combines convolutional and transformer layers with efficient local attention modules. This design improves efficiency and accuracy on benchmarks such as GTSRB and BTSD. However, despite architectural optimizations, E-MobileViT still outputs a prediction for every input, without mechanisms to handle low-confidence cases. Overall, ViT represents a promising paradigm for traffic sign recognition, offering improved accuracy, interpretability, and robustness compared to traditional CNNs. However, to enable large-scale deployment in real-world autonomous systems, further optimization in terms of computational efficiency, model generalization, and safety-critical interpretability is required.

Given these limitations, safety requires not only accurate classification but also mechanisms that prevent unreliable predictions from influencing vehicle decisions. Fail-controlled systems, widely applied in avionics and railways, offer this capability by discarding uncertain outputs. In the context of machine learning, fail-controlled classifiers (FCCs) have gained traction as a means to mitigate the risks associated with misclassification. A recent work [31] formalized the concept of FCCs, proposing a framework that combines self-checking mechanisms, input/output processors, and safety wrappers to enable FCC behavior. [32] Empirically assess the properness by analyzing the distribution of binary classifier misclassifications instead of simply counting misclassifications. These studies highlight the potential of FCCs to improve the safety and reliability of autonomous systems. However, further research is needed to adapt these approaches to specific domains, such as traffic sign recognition. [33] revisited physical-world adversarial attacks on commercial TSR systems, showing that real deployments remain vulnerable even when benchmark performance is high. Their findings highlight the gap between experimental accuracy and system-level safety. Also, human drivers often rely on subjective risk assessments when navigating uncertain environments, and similar safety-driven considerations must be embedded into autonomous systems [34]. Inspired by the success of transformers in NLP, [35] proposed the ViT, which uses self-attention mechanisms to segment an image into patches and process them as tokens. ViT and its variants have demonstrated impressive results on various vision benchmarks [36,37]. Nonetheless, they assume equal importance for all input patches and do not account for context-dependent relevance, a limitation that is particularly impactful in safety-critical tasks such as TSR, where focusing on meaningful regions is essential. A comprehensive comparison of the techniques discussed in this section is shown in Table 1 below:

## 4 Materials and methods

The proposed CViT comprises three major phases. The first phase is the preprocessing phase, in which the input traffic sign images are divided into different patches. Each patch is then flattened, and positional embeddings are added to each flattened patch. This phase results in a complete class token input sequence. The feature extraction phase is the second

**Table 1. Comparison of some existing approaches to Traffic Sign Recognition (TSR).**

| Approach | Strengths | Limitations | Relation to Our Work |
|---|---|---|---|
| CNN-based TSR [10,11,13,14] | High accuracy on benchmarks (e.g., GTSRB); effective feature extraction | Sensitive to occlusion, illumination, and adversarial noise; limited global context | Our CViT addresses the global context via transformers |
| Hybrid CNN + Data Augmentation [14–16] | Improved robustness; lower overfitting | Still prone to misclassification; ignores safety risks | CViT incorporates adaptive attention and fail-safe control |
| ViT [20–22,24,25,36,37,38] | Capture global dependencies; strong performance on large datasets | Require large pretraining; treat all patches equally; high computation cost | CViT introduces conditional attention and adaptive Q/K/V to reduce redundancy |
| Hybrid CNN–ViT models [26,27] | Combine local + global features; better generalization | Complexity: increased computation | CViT balances complexity with conditional gating |
| Fail-Controlled Classifiers (FCCs) [31,32] | Reduce unsafe outputs; enhance reliability | Mostly conceptual; limited application to TSR | CViT integrates FCC principles directly into the ViT architecture |

phase, in which the patch embeddings sequence is processed through multiple conditional transformer layers. Each layer applies a multi-head attention (MHA) mechanism that extracts rich context-aware features. The last phase is the classification phase, in which the class tokens extracted through the second phase are passed through a fully connected layer to produce the final prediction of the traffic sign images. The proposed methodology is shown in Fig 1 below:

The detailed steps involved in each phase of the proposed methodology are discussed in the following section.

### 4.1 Data preprocessing.

In the data processing phase, the CViT prepares the input raw traffic sign image by dividing it into small patches, converting them into embeddings, and then adding positional information to these embeddings. This step transforms the traffic sign image into a structured sequence that the CViT can process easily. Fig 2 shows the steps performed in this phase, and they are described in detail as follows:

**4.1.1 Patch Extraction.** The first step in the data preprocessing phase is to collect a dataset of traffic sign images, each labeled with the correct class. These images are first resized to a fixed resolution as $H * W$, e.g., 224 × 224 pixels. Each resized image $I \in \mathbb{R}^{H*W*C}$, where H and W denote height and width, and C denotes the number of channels (e.g., C = 3 for RGB images). The total number of patches extracted from the image can be calculated as follows:

$$N = \frac{HW}{P^2}$$

(1)

Then, each patch $x_i$ $for (i = 1, 2, \ldots, N)$ are flattened into a vector of size $P^2 \cdot C$.

**4.1.2 Patch Embedding and Positional Encoding.** In this second step, for obtaining patch embeddings $e_i \in \mathbb{R}^D$, each flattened patch is projected via a trainable linear layer into a higher-dimensional embedding space. Then, positional encodings are added so that the spatial relationship between patches can be retained. The patch embedding can be represented as follows:

$$e_i = W_E \cdot x_i + b_E, \quad e_i \in \mathbb{R}^D$$

(2)

Where $W_E \in \mathbb{R}^{D*\ (P^2 \cdot C)}$ and $b_E \in \mathbb{R}^D$ are learnable parameters, and $D$ is the embedding dimension. The positional encoding for the patch can be represented as follows:

$$z_i = e_i + p_i$$

(3)

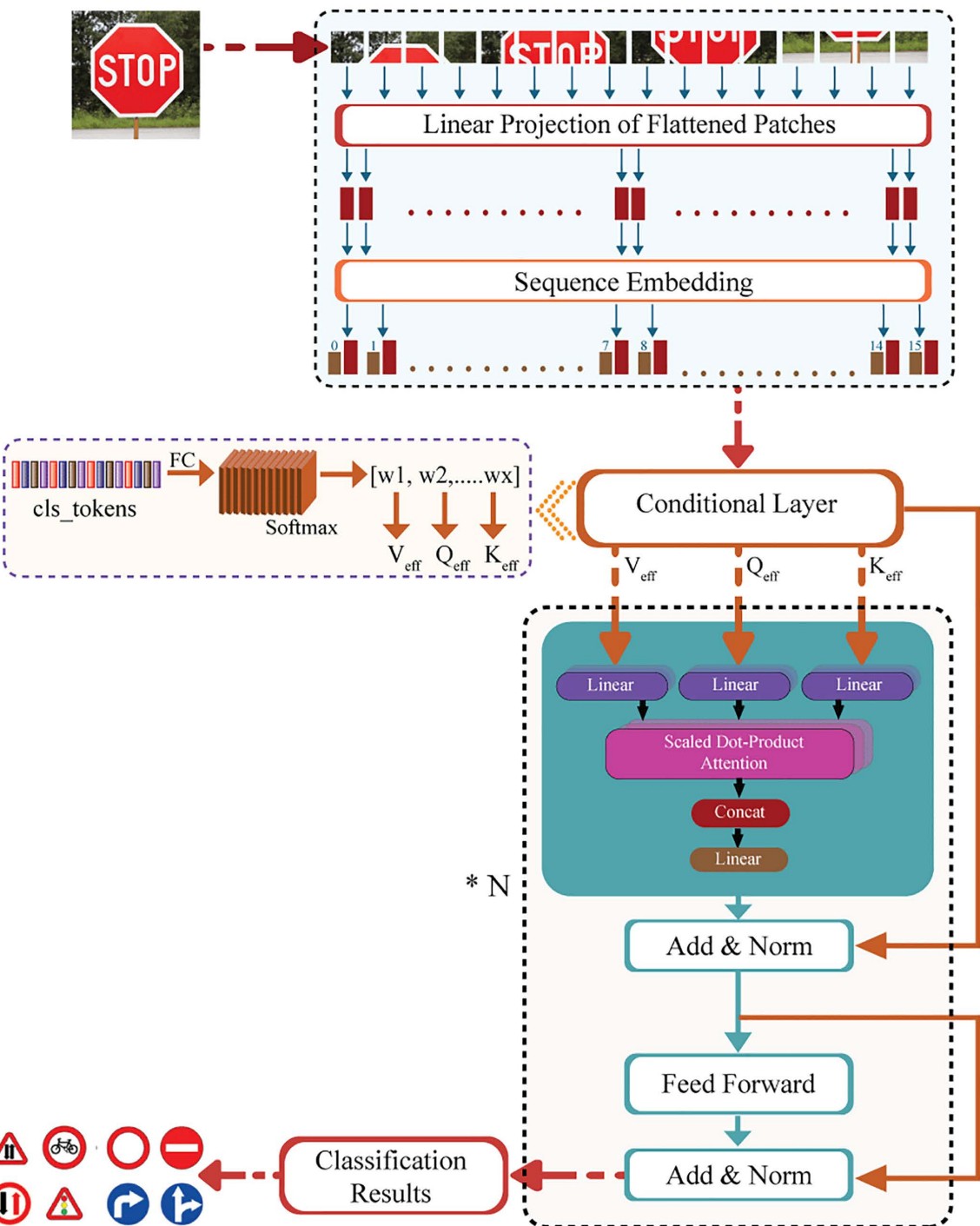

**Fig 1. Proposed Conditional Visual Transformer (CViT) Model.**

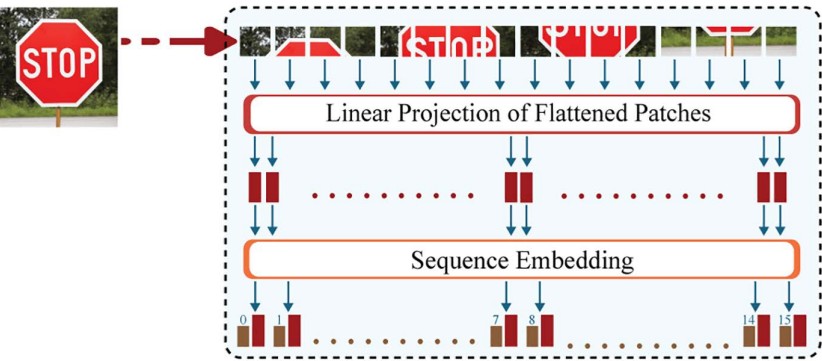

**Fig 2. Preprocessing Phase of the Proposed CViT Model.**

Where $p_i \in \mathbb{R}^D$ represents the positional encoding of patch i.

**4.1.3 Forming the Token Sequence.** The next step of the preprocessing phase is to generate learnable class tokens $e_{cls} \in \mathbb{R}^D$ to represent global information. Learnable positional embeddings $p_{cls}$ are added to this class token. The complete input sequence is then formed by combining the class token with the patches' tokens. The class token formation of the final token sequence $Z = [z_{cls}, z_1, z_2, ...., z_N]$ can be represented as follows:

$$z_{cls} = e_{cls} + p_{cls} \tag{4}$$

This sequence $Z$ is the input to the conditional transformer encoder.

## 4.2 Feature extraction through conditional transformer layer

The second phase is the feature extraction through the conditional transformer layer. This phase focuses on extracting contextual features from the token sequence using a conditional attention mechanism. A gating 00compute the adaptive Q, K, and V representations. These dynamic projections help the model attend differently based on the type of traffic sign image input. Phase 2 of the proposed methodology is shown in Fig 3 and each step is described in the next sub-sections.

**4.2.1 Gating network (Global features to weights).** In the first step, the global features are processed through a fully connected layer with softmax activation to generate the weights. The computed weights $w$ can be represented as follows:

$$w = softmax(W_g \cdot FC(GlobalFeature) + b_g) \tag{5}$$

Where

- $W_g \in \mathbb{R}^{x*d}$ is the learnable weight matrix of the FC layer,

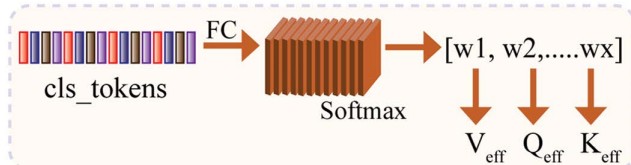

**Fig 3. Gating Network for Conditional Q, K, and V Projection.**

- $b_g \in \mathbb{R}^x$ the learnable bias vector that shifts the FC layer outputs before softmax,

- $w = [w_1, w_2, \ldots, w_x]$ is the resulting normalized weight vector such that $\sum_i w_i = 1$.

**4.2.2 Conditional Q, K, V projection.** In the proposed CViT, adaptive Q, K, and V matrices are computed through a weighted sum over several expert matrices instead of fixed projection matrices. The effective projections are represented in Eq (6) as follows:

$$Q_{eff} = \sum_{i=1}^{x} w_i Q_i, \ K_{eff} = \sum_{i=1}^{x} w_i K_i, \ V_{eff} = \sum_{i=1}^{x} w_i V_i \tag{6}$$

Where $Q_i$, $K_i$, $V_i$ are the expert projection matrices. This allows the conditional layer to integrate traffic sign-relevant features with the current target embeddings (the patch embeddings).

**4.2.3 Conditional Multi-Head Attention (MHA).** In the next step, adaptive $Q_{eff}$, $K_{eff}$, $V_{eff}$ matrices are fed into the MHA mechanism to compute the attention score and output representation. The self-attention computation can be described as follows:

$$Attention\,(Q_{eff}, K_{eff}, V_{eff}) = Softmax\left(\frac{Q_{eff} \cdot K_{eff}^T}{\sqrt{d_k}}\right) \cdot V_{eff} \tag{7}$$

Where $d_k$ is the dimension of the key vectors. Then, a multi-head mechanism is applied in which the projections are split into multiple heads, and each head computes its attention. The block inside the dotted region is repeated $N$ times, where $N$ denotes the number of stacked transformer encoder layers. The outputs are then concatenated and linearly projected back to the original embedding dimension. The conditional Multi-Head attention with residual connections and feed-forward network is depicted in Fig 4.

**4.2.4 Residual Connections and Feed-Forward Network (FFN).** After applying the attention mechanism, a residual connection and layer normalization are applied, followed by a feed-forward network to refine the traffic sign features further. The Residual & Normalization after Attention, Feed-Forward Network (FFN), and Residual & Normalization after FFN follow the standard transformer encoder formulation [38,39], which can be mathematically represented as follows:

$$Z' = LayerNorm(Z + Attention(Z)) \tag{8}$$

$$FFN\,(Z') = \sigma\,(W_1 Z' + b_1)\,W_2 + b_2, \ where \ \sigma \ is \ activation \ function \tag{9}$$

$$Z'' = LayerNorm(Z' + FFN\,(Z')) \tag{10}$$

The output from repeated blocks of conditional attention and FFN forms the final encoded representation of the traffic sign image. This contextual representation of the input traffic sign image encodes the dynamic attention-based relationships among patches and is further passed for classification.

**4.3 Classification of traffic signs.** In the final phase of CViT, the refined global feature representation is now used to classify traffic signs into different categories. This final phase condenses the transformer encoder output into a decision through the classifier. In the first step of this phase, the final feature is extracted from the output of CViT. Then, the extracted global feature is fed into a classification head, which is a fully connected layer that maps it to class logits. The mapping of logic $\hat{y}$ and prediction $P(class\,|image)$ can be represented as follows:

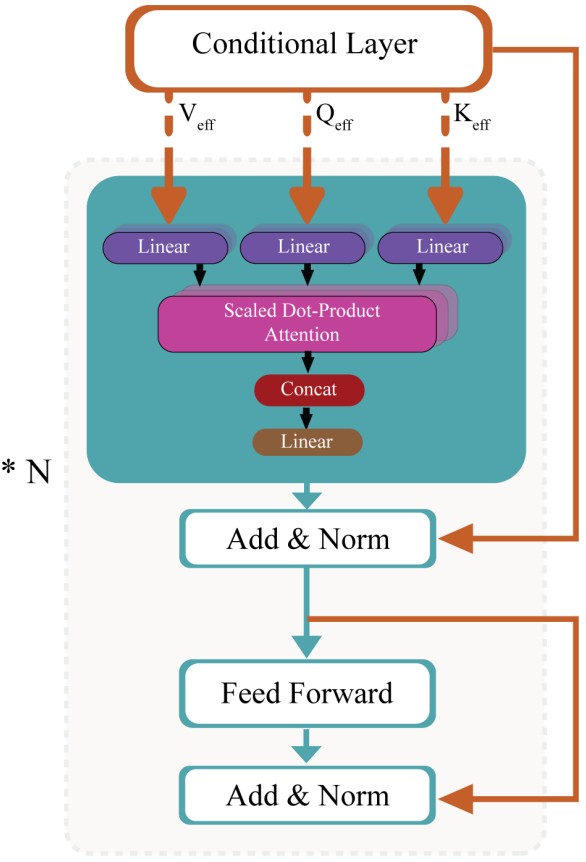

**Fig 4. Conditional multi-head attention, residual connections, and feed-forward network.**

$$\hat{y} = W_c \cdot Z_{final} + b_c \tag{11}$$

$$P(class \mid image) = Softmax(\hat{y}) \tag{12}$$

Here, $(\hat{y})$ are logits and the $Softmax(\hat{y})$ produces class probabilities for the traffic sign. The proposed CViT's complete algorithm can be described as follows:

```
input: Traffic sign image (X)
output: Class label, y
begin
1.Pre-processing Phase
    i) The input image X is resized to a fixed resolution (e.g., 224×224).
    ii) Divide image X into N non-overlapping patches of size P×P.
    iii) Flatten each patch xi into a vector of size P²·C, where C = number of channels.
    iv) Project each patch xi into embedding space: ei = WE·xi+bE, ei ∈ ℝD
    v) Add positional encoding pi to each embedding to retain spatial information: zi= ei+pi
    vi) Generate a learnable class token ecls ∈ ℝD and add its positional encoding: zcls= ecls + pcls
    vii) Form the complete input sequence to the transformer: Z=[zcls, z1, z2,.....,zN}
2.Feature Extraction Phase using Conditional Transformer
```

```
i. Compute the global features from the input sequence Z.
ii. Pass global features through a gating network with SoftMax to obtain weights
```
$w = [w_1, w_2, \ldots, w_x]$: $w = softmax(W_g \cdot FC(GlobalFeature) + b_g)$

```
iii. Compute adaptive projections using the weighted sum of expert matrices:
```

$$Q_{eff} = \sum_{i=1}^{x} w_i Q_i \ , \ K_{eff} = \sum_{i=1}^{x} w_i K_i, \ V_{eff} = \sum_{i=1}^{x} w_i V_i$$

```
iv. Applying Conditional MHA:
```

$$Attention(Q_{eff}, K_{eff}, V_{eff}) = Softmax\left(\frac{Q_{eff} \cdot K_{eff}^T}{\sqrt{d_k}}\right) \cdot V_{eff}$$

```
v. Add the residual connection and layer normalization:
```
$Z' = LayerNorm(Z + Attention(Z))$

```
vi. Pass through the Feed-Forward Network:
```
$FFN\left(Z'\right) = \sigma\left(W_1 Z' + b_1\right) W_2 + b_2$

```
vii. Apply the second residual connection and normalization:
```
$Z'' = LayerNorm\left(Z' + FFN\left(Z'\right)\right)$

```
viii. Steps (b)-(g) are repeated through multiple Conditional Transformer blocks.
3.Classification Phase
    i) Extract the final encoded class token Z_final from the output of the transformer.
    ii) Pass it through a fully connected classification head to generate logits:
```
$\hat{y} = W_c \cdot Z_{final} + b_c$

```
    iii) Applying SoftMax to obtain class probabilities and predict label:
```
$P(class \mid image) = Softmax(\hat{y})$

```
End
```

The fail-control strategy is also integrated as a post-classification decision layer in the CViT Model. Following the model's output probabilities, a confidence threshold is applied to identify predictions with insufficient certainty. If the maximum softmax probability falls below this predefined threshold $p_{thr} = 0.8$, the output is intentionally omitted to avoid risking a low-confidence and potentially incorrect classification. The fail-control mechanism is shown in Fig 5. This rejection mechanism is inspired by previous work on fail-safe classifiers, particularly those discussed in safety-critical machine learning frameworks [31,32,35–37]. By treating omission as a valid and intentional outcome, the system can balance accuracy and operational safety, especially in real-world deployments where uncertainty cannot be ignored.

The classifier generates an initial class prediction for input data (traffic sign image). This prediction is passed to an output checker, which evaluates whether the result is "trustable" based on confidence thresholds and fail-control metrics. If trustable, the prediction is accepted as part of the final classification results. If not trustable, the prediction is discarded to avoid misleading or unsafe outputs. The mechanism integrates metrics from Table 7 to decide on trustworthiness. For instance, α measures how many predictions are accepted, ε tracks error, while φ quantifies critical omissions. This layered decision process improves robustness by ensuring only reliable predictions are used in downstream applications such as autonomous driving.

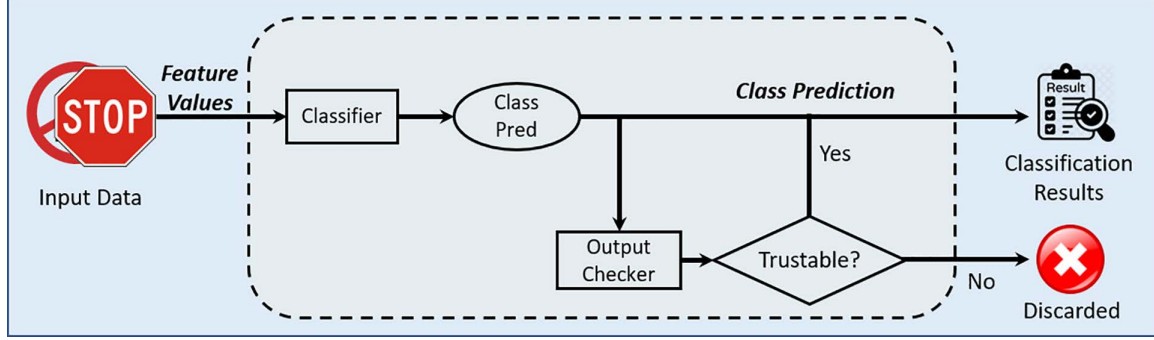

**Fig 5. Fail-control mechanism for classification.**

## 5 Experimental setup and results

The experimental evaluation of the proposed CViT was experimentally evaluated on the German Traffic Sign Recognition Benchmark (GTSRB) dataset, which comprises over 50,000 images of 43 distinct traffic sign classes. Images are resized to 32×32 pixels and augmented using random horizontal flips, rotations, and color jitter for the training set, while standard resizing is used for validation and testing. The dataset was divided into training, validation, and testing subsets using a 80:10:10 split, ensuring that class distribution was maintained across subsets. A custom PyTorch Dataset class reads the image paths and labels from the CSV files. A *WeightedRandomSampler* function is applied during training to address class imbalance. The CViT architecture includes patch embedding, a gating network, and multiple transformer encoder layers with conditional MHA using expert-based query, key, and value projections. The proposed model comprised approximately 12.3 million trainable parameters, reflecting its computational complexity and capacity to capture fine-grained patterns in traffic sign images. The model is trained for 50 epochs using the Adam optimizer and cross-entropy loss. To avoid overfitting and ensure convergence, early stopping and learning rate scheduling were employed. The model was trained using the PyTorch deep learning framework on a machine equipped with an NVIDIA RTX GPU, 16 GB RAM, and an AMD RYZEN.

For evaluating the performance of the proposed CViT, different experimental setups were conducted using a publicly available GTSRB dataset, which has multi-class traffic signs, with approximately 43 distinct traffic sign categories. The classification performance is assessed using different performance metrics, such as accuracy, precision, recall, and F1-score, for each class. In addition, macro, micro, and weighted averages are measured to demonstrate the model's overall performance.

Initial experiments were conducted using a simple ViT architecture to evaluate the effectiveness of the proposed CViT. This baseline model served as a reference to assess how well the conditional attention mechanism improves performance. The same dataset and evaluation metrics were then applied to the CViT model. These results are discussed one by one as follows:

### 5.1 Experimental results using simple Vision Transformer (ViT)

First, the traffic sign images are passed to a simple ViT model. Although the simple ViT achieved an overall accuracy of 94.25%, the class-wise results show that these results can also be improved. The performance measures of the top 5 best-performing classes and the top 5 least-performing classes are shown in Table 2. T21, T2, and T41 exhibited the

**Table 2. Top 5 Best-performing and top 5 least-performing classes for the simple ViT.**

| Class | Accuracy | Precision | Recall | F1 Score |
|-------|----------|-----------|--------|----------|
| Top 5 Best-Performing Classes | | | | |
| T8 | 98.61 | 95 | 88 | 90 |
| T6 | 96.79 | 88 | 81 | 83 |
| T17 | 97.27 | 81 | 82 | 81 |
| T12 | 96.07 | 88 | 83 | 86 |
| T29 | 95.85 | 82 | 76 | 79 |
| Top 5 Least-Performing Classes | | | | |
| T21 | 91.60 | 61 | 59 | 59 |
| T2 | 92.27 | 60 | 63 | 61 |
| T41 | 93.74 | 76 | 65 | 71 |
| T5 | 94.00 | 77 | 69 | 73 |
| T10 | 94.32 | 67 | 87 | 73 |

lowest accuracy values, with T21 being the most misclassified, achieving an accuracy of 91.60%. Here, T1–T43 denote the 43 traffic sign categories defined in the GTSRB dataset (e.g., T1 = Speed Limit 20, T2 = Speed Limit 50, T21 = Keep Right, T41 = End of No Passing, etc.). Compared with the best-performing class T8 (98.61%), T2 shows a 7.01% drop in accuracy, indicating more frequent confusion during classification.

T21 and T2 also have the lowest precision and recall scores (around 60%–63%), indicating high misclassification rates in both directions. The recall for T21 (59%) is the lowest across all classes, indicating that the model frequently fails to correctly identify true instances of this class. Only the five best-performing and five least-performing classes are shown to highlight extremes in performance. The micro average, macro average, and weighted average are also measured, as shown in Table 3 below.

### 5.2 Conditional Vision Transformer (CViT) Results

After collecting the classification result of traffic sign images through a simple ViT model, the CViT model is evaluated. To ensure a fair comparison, both models were trained and evaluated on the same dataset under similar conditions. The results reveal significant improvements in overall accuracy, per-class stability, and generalizability with CViT. The performance measures of the top 5 best-performing classes and the top 5 least-performing classes are shown in Table 4. The CViT significantly outperformed the simple ViT, achieving an overall accuracy of 99.87%. Its per-class consistency and high precision-recall values indicate a robust classification capability. T21, T2, and T41 still exhibit slightly lower accuracy than other classes, with T1 achieving the lowest accuracy of 99.27%.

**Takeaway 2:** The model achieves outstanding performance on the GTSRB dataset with 99.87% accuracy, significantly outperforming the baseline ViT and recent approaches.

The accuracy difference between T21 and the top-performing class T12 (99.89%) is just 0.62%, indicating remarkable consistency across classes. While all precision and recall values are above 92%, T21's recall (92%) is still the lowest, suggesting room for improvement in identifying true positives for this class. Unlike

**Table 3. Micro average, macro average, and weighted average of simple ViT.**

| Metric | Micro Avg | Macro Avg | Weighted Avg |
|---|---|---|---|
| Precision | 94.25 | 75.7 | 75.7 |
| Recall | 94.25 | 75.5 | 75.5 |
| F1 Score | 94.25 | 75.6 | 75.6 |

**Table 4. Top 5 best-performing and top 5 least-performing classes for CViT.**

| Class | Accuracy | Precision | Recall | F1 Score |
|---|---|---|---|---|
| Top 5 Best-Performing Classes | | | | |
| T8 | 99.89 | 99.0 | 98.8 | 99.0 |
| T6 | 99.80 | 99.8 | 97.0 | 99.0 |
| T17 | 99.73 | 97.0 | 98.0 | 98.0 |
| T12 | 99.69 | 98.9 | 98.2 | 98.0 |
| T29 | 99.62 | 99.0 | 96.0 | 98.0 |
| Top 5 Least-Performing Classes | | | | |
| T21 | 99.27 | 92.0 | 92.0 | 92.0 |
| T2 | 98.97 | 94.0 | 96.0 | 96.0 |
| T41 | 99.56 | 93.0 | 95.0 | 95.0 |
| T10 | 99.50 | 94.0 | 95.0 | 96.0 |
| T4 | 99.25 | 96.0 | 97.0 | 97.0 |

**Table 5. Micro average, macro average, and weighted average of CViT.**

| Metric | Micro Avg | Macro Avg | Weighted Avg |
|---|---|---|---|
| Precision | 94.1 | 99.07 | 94.1 |
| Recall | 94.3 | 99.07 | 94.3 |
| F1 Score | 94.2 | 99.07 | 94.2 |

**Table 6. Comparison of the performance measures of simple ViT and CViT.**

| Metric/ Class | Simple ViT | CViT |
|---|---|---|
| Overall Accuracy | 94.25% | 99.87% |
| Most Misclassified Class | T21 (91.60%) | T21 (99.27%) |
| Top Performing Class | T8 (98.61%) | T8 (99.89%) |
| Accuracy Gap (Lowest vs. Best) | 7.01% (T2 vs. T8) | 0.62% (T21 vs. T12) |
| Lowest Recall Class | T21 (59%) | T21 (92%) |
| Precision (Micro/ Macro/ Wtd Avg) | 94.25%/75.7%/75.7% | 94.1%/99.07%/94.1% |
| Recall (Micro/ Macro/ Wtd Avg) | 94.25%/75.5%/75.5% | 94.3%/99.07%/94.3% |
| F1 Score (Micro/ Macro/ Wtd Avg) | 94.25%/75.6%/75.6% | 94.2%/99.07%/94.2% |
| Top 5 Best Classes – Avg F1 | ~83.8% | ~98.4% |
| Top 5 Worst Classes – Avg F1 | ~67.4% | ~95.2% |

the Simple ViT, the F1 scores for all classes are tightly grouped between 92% and 99%, indicating a balanced prediction quality. Only the five best-performing and five least-performing classes are shown to highlight extremes in performance. The micro average, macro average, and weighted average are also measured, as shown in Table 5 below:

The comparison of simple ViT and CViT is shown in Table 6. The comparison in the table shows that the proposed CViT clearly outperforms the simple ViT. CViT achieved a higher overall accuracy (99.87% vs. 94.25%) and improved recall for the most misclassified class (T21) from 59% to 92%. It also narrowed the accuracy gap between the best- and worst-performing classes from 7.01% to only 0.62%, reflecting more balanced learning across all categories. Moreover, CViT consistently delivered stronger precision, recall, and F1-scores, and raised the performance of even the weakest classes (from ~67.4% to ~95.2% in average F1). These results highlight that CViT not only achieves top-level accuracy but also handles challenging cases more effectively, making it highly reliable for autonomous driving applications.

**Takeaway 3:** CViT demonstrates the potential for fail-controlled, interpretable, and highly accurate models in autonomous driving and other safety-critical applications.

To validate the effectiveness of the proposed CViT, a comparative analysis is performed on the GTSRB dataset using some recent approaches. Fig 6 compares the performance of four recent models [40–42] along with the Simple ViT and the proposed CViT. Although these existing models achieved notable accuracies ranging from 98.41% to 99.66%, the CViT outperforms them all, attaining 99.87% accuracy. Specifically [41] shows the highest accuracy among the models mentioned in the table, achieving 99.66%, followed by [42] at 98.5%, and [40] at 98.41%. However, the Simple ViT baseline achieved an accuracy of 94.25%. In contrast, the CViT significantly boosts the classification accuracy by 5.62% over the Simple ViT, and by 0.21% over [41]. These results highlight the robustness and reliability of the proposed CViT for better classification of traffic signs with different types of images by refining the feature extraction process. The key reason for this improvement is CViT's adaptive attention mechanism, which dynamically adjusts Q, K, and V projections based on input variations, unlike fixed-attention models that struggle with occlusions, illumination changes, and diverse traffic sign appearances.

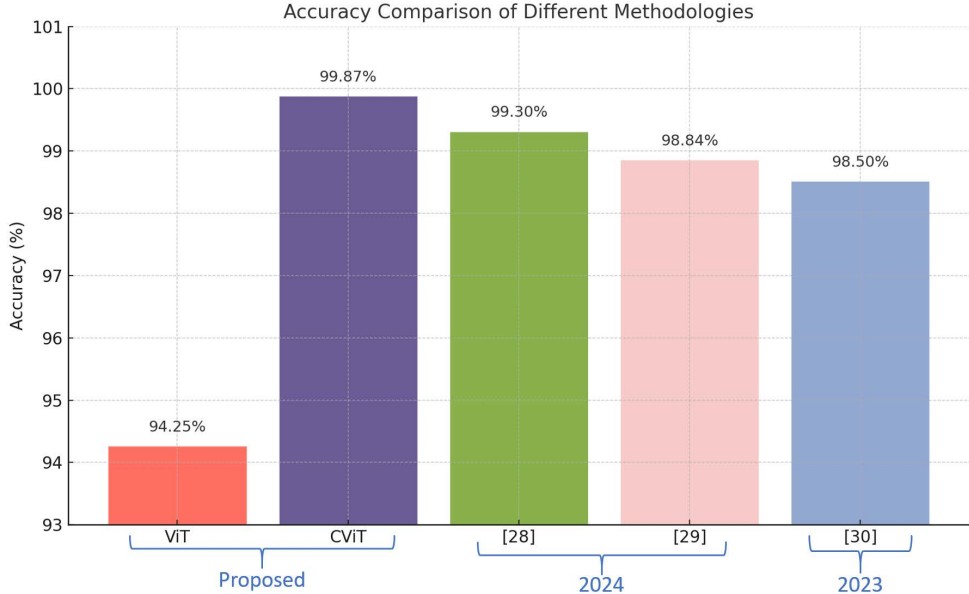

**Fig 6. Graphical representation of comparison with existing ViT-based techniques.**

## 5.3 Fail control mechanism on the CViT Model

The fail-control strategy is embedded in the CViT output layer through a threshold-based confidence filter. After the model generates class probabilities, predictions with softmax scores below the confidence threshold $p_{thr} = 0.8$ are rejected. These inputs are labeled as "non-trustable" and omitted from classification. To assess the effectiveness of a fail-controlled mechanism, several key evaluation metrics are used, which are shown in Table 7 below:

These metrics provide a clear picture of the trade-off between predictive accuracy and system reliability under fail-controlled operation. Table 8 summarizes the results after applying the fail-control mechanism to the classification results. This table shows the behavior of the model under both standard and fail-controlled conditions.

The base model achieved an overall accuracy of 82.1%. However, by activating the fail-control mechanism, the system reduced its error rate from 17.9% to 7.9%, resulting in a relative error reduction of 55.9%. Additionally, approximately 54.5% of misclassifications were effectively omitted, demonstrating the rejection strategy's ability to prevent

**Table 7. Key evaluation metrices used for evaluating fail control mechanism.**

| Parameter | Description |
|---|---|
| $\alpha$ | Overall Accuracy: Proportion of all correct predictions made by the base model |
| $\varepsilon$ | Overall Error Rate: Proportion of incorrect predictions across the entire dataset |
| $\varphi$ | Omission Rate: Fraction of inputs for which the model abstained from making a prediction |
| $\alpha_w$ | Controlled Accuracy: Accuracy computed only on the subset of predictions that were not omitted |
| $\varepsilon_w$ | Controlled Error Rate: Error rate among the accepted predictions after fail-control |
| $\varphi_c$ | Critical Omissions: Correct predictions that were omitted unnecessarily due to low confidence |
| $\varphi_m$ | Misclassification Omissions: Incorrect predictions that were successfully caught and omitted |
| $\varepsilon_{gain}$ | Error Reduction: Relative decrease in error achieved by the rejection mechanism |
| $\varphi_{m\_ratio}$ | Misclassification Capture Ratio: The proportion of all misclassifications that were effectively removed through omission |

**Table 8. Results of applying the fail control mechanism to the CViT model: α (Acceptance Rate), ε (Error Rate), φ (Critical Omissions), α_w (Weighted Acceptance Rate), ε_w (Weighted Error Rate), φ_c (Critical Omissions – Corrected), φ_m (Missed Predictions), ε_gain (Error Reduction), φ_m_ratio (Missed Prediction Ratio).**

| Metric | Value |
| --- | --- |
| $\alpha$ | 0.821 |
| $\varepsilon$ | 0.179 |
| $\varphi$ | 0.183 |
| $\alpha_w$ | 0.738 |
| $\varepsilon_w$ | 0.079 |
| $\varphi_c$ | 0.083 |
| $\varphi_m$ | 0.099 |
| $\varepsilon_{gain}$ | 0.559 |
| $\varphi_{m\_ratio}$ | 0.545 |

incorrect predictions from reaching the decision stage. Although some correct predictions were also omitted (8.3%), this trade-off is considered acceptable in safety-critical contexts. The slight reduction in accuracy among accepted predictions (from 82.1% to 73.8%) reflects a more cautious classification strategy, which prioritizes correctness over coverage. The FC-CViT model demonstrates the benefits of integrating fail-control into high-performing architectures. Although its overall accuracy ($\alpha$ = 82.1%) is lower due to deliberate omissions, its controlled accuracy on accepted predictions ($\alpha_w$) remains high at 73.8%. More importantly, the rejection mechanism reduces the error rate by 55.9%, and over half (54.5%) of potential misclassifications are effectively eliminated. These improvements are particularly impactful in real-world applications where the cost of a wrong prediction is far greater than that of no prediction.

## 6 Conclusion

In this study, a CViT model is proposed, comprising conditional query-key-value (Q, K, V) projections and attention-based mechanisms for robust and accurate traffic sign classification for autonomous vehicles. It also uses an adaptive gating network for better projection. The proposed method is evaluated on the publicly available GTSRB dataset, which is a multi-class dataset comprising 43 unique traffic sign classes. After experimenting on the dataset using the CViT, the proposed model significantly outperformed the baseline ViT model. It achieved an overall accuracy of 99.87%, precision of 99.07%, recall of 99.07%, and F1-score of 99.07%, demonstrating its effectiveness across all traffic sign classes. The proposed method is also compared with recent state-of-the-art approaches, and the results confirm that CViT not only provides superior accuracy but also maintains balanced precision, recall, and F1 performance. The findings highlight the importance of adaptive attention-based mechanisms in ViT architectures for tasks such as object detection and traffic sign recognition. This work opens a new direction for adopting this mechanism in fail-controlled, interpretable, and highly accurate detection and recognition for real-world autonomous driving systems and other safety-critical applications.

In the future, this work can be extended by incorporating fail-control mechanisms more explicitly to reduce misclassification in safety-critical environments, integrating explainable AI (XAI) techniques for better interpretability of decisions, and adapting the CViT framework for real-time deployment on embedded systems in autonomous vehicles. Further, adversarial testing frameworks such as critical scenario generation can also provide a promising avenue to evaluate the robustness of fail-controlled recognition models under safety-critical conditions [43].

## Supporting information

**S1 File. Supporting_information_file.**
(ZIP)

## Author contributions

**Conceptualization:** Isra Naz, Ali Tahir, Rabia Saleem.

**Data curation:** Isra Naz.

**Formal analysis:** Isra Naz, Rabia Saleem.

**Investigation:** Mutaz Elradi S. Saeed.

**Methodology:** Isra Naz, Jamal Hussain Shah, Mahatma Reddy Marri.

**Project administration:** Ali Tahir.

**Resources:** Mutaz Elradi S. Saeed.

**Software:** Isra Naz, Jamal Hussain Shah, Mahatma Reddy Marri.

**Supervision:** Jamal Hussain Shah, Rabia Saleem.

**Validation:** Mutaz Elradi S. Saeed.

**Visualization:** Jamal Hussain Shah.

**Writing – original draft:** Isra Naz.

**Writing – review & editing:** Ali Tahir, Rabia Saleem, Mahatma Reddy Marri.

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
