## [Decision Letter · Decision Letter 0]

3 Sep 2025

Thank you for submitting your manuscript to PLOS ONE. After careful consideration, we feel that it has merit but does not fully meet PLOS ONE’s publication criteria as it currently stands. Therefore, we invite you to submit a revised version of the manuscript that addresses the points raised during the review process.

Authors are advised to revise the manuscript as per the reviewers' comments. 

We look forward to receiving your revised manuscript.

Kind regards,

Mudassir Khan, Ph.D

Academic Editor

PLOS ONE

Journal Requirements:

Additional Editor Comments:

Authors are advised to revise the manuscript as per the reviewers' comments.

Reviewers' comments:

Reviewer's Responses to Questions

**Comments to the Author**

1. Is the manuscript technically sound, and do the data support the conclusions?

Reviewer #1: Yes

Reviewer #2: Yes

2. Has the statistical analysis been performed appropriately and rigorously?

Reviewer #1: Yes

Reviewer #2: Yes

3. Have the authors made all data underlying the findings in their manuscript fully available?

Reviewer #1: Yes

Reviewer #2: Yes

4. Is the manuscript presented in an intelligible fashion and written in standard English?

Reviewer #1: Yes

Reviewer #2: Yes

Reviewer #1: The manuscript addresses a highly relevant challenge of traffic sign recognition for autonomous vehicles, proposing a Conditional Visual Transformer (CViT) with a fail-control mechanism. However, improvements in writing clarity, figure quality, formatting consistency, and critical discussion are required.

1- The sentence structure in abstract need to be improve i-e. “While issues related to object recognition, such as traffic sign recognition… “ is too lengthy and contains overly technical descriptions. Abstract should instead briefly state the problem, the novelty of the approach, key results, and implications.

a. Phrases like “gratuitous” are misused and should be replaced with clearer academic wording.

2- The introduction is informative but contains repetition (e.g., multiple mentions of misclassification risks and safety-critical concerns). Restructuring into a concise flow would improve readability.

a. Sentence like “…a misplaced label can be immensely disastrous will make this more problematic” is grammatically awkward and should be revised for clarity.

3- The listed contributions are significant; however, they should be rewritten more concisely and distinctly in a separate heading, avoiding overlap with methodology descriptions.

a. Also, explain the third contribution, “Proposed CViT incorporates a specialized attention module… “What is meant by specialized attention module here? And how is it specialized?

4- The methodology is presented in detail with equations and figures, which is commendable. However:

a. Several equations (e.g., Eq. (1), Eq. (2)–(4)) lack variable definitions. Every parameter (e.g., P, C, d) should be explicitly explained, as this can confuse readers unfamiliar with the notation.

b. In Eq. (5) bg is not described, what is bg and what is its use in Eq. 5?

c. In 3.1.1 “Then each patch x_i for(i=1,2,…,N) are flattened into a vector of size P^2∙C, where C is the number of channels, for example, for RGB images C=3.” is mentioned but again repeated at the end of 3.1.2. It needs explanation, why is it repeated again?

5. Several grammar issues (e.g., “Figure 6 demonstrate” should be “demonstrates”). The manuscript needs a thorough language polish.

6. Ensure that acronyms (e.g., GAP, FFN, MHA) are introduced at first use and used consistently throughout.

7. Equation numbering is inconsistent, like in Eq. 1 to Eq. 10, parentheses “()” are used, but in Eq. 11-12, square braces “[ ]” are used.

8. In Section 3.1.1, the phrase “as e.g., 224×224” is unpolished. Suggested edit: “e.g., resized to 224×224 pixels.”

9. Reference formatting is inconsistent: some include DOIs, others do not. Standardize per journal guidelines.

Reviewer #2: The manuscript addresses an important and timely challenge in traffic sign recognition for autonomous vehicles by introducing a Conditional Visual Transformer (CViT) with an integrated fail-control mechanism.

The proposed approach is technically relevant and well-motivated. However, the manuscript would benefit from improved writing clarity, figure presentation, and formatting consistency, as well as a more comprehensive critical discussion of the results and limitations to align with the standards of the journal before final acceptance.

1) (a) The related work section is extensive; however, the flow is occasionally interrupted by abrupt transitions. In addition, it lacks critical analysis and a discussion of the limitations of prior studies. It is also unclear why ViT is specifically preferred or emphasized.

(b) Furthermore, please add a new table in the related work section that summarizes the pros and cons of previous methods and clearly illustrates how your contribution relates to them and provides improvements.

2) In Figure 4, what is *N? is it mistaken? if not, please properly explain its working or importance in Figure 4.

3) In 3.2.4, “The Residual & Normalization after Attention, Feed-Forward Network (FFN), and Residual & Normalization after FFN are mathematically represented …” are these the modified equations or just the basic ones? If modified, then please explain them. And if these are just the same as the basic Vision transformer, then please include references.

4) The algorithm of the proposed Conditional Vision Transformer is good and well describes the whole methodology. The fail-control mechanism is a key strength, but its explanation is somewhat limited, and it is not discussed in the abstract.

(a) Table 7 includes valuable metrics, but abbreviations (e.g., “Critical Omissions,” “Error Reduction”) should be explained in captions.

(b) Figure 5 needs to be explained properly with details of metrics and parameters.

5) There are typos in the experimental section may be like “color jitter”, “WeightedRandomSampler”. It also misses the details about the training-testing ratio for the experiments.

(a) Fail Control Mechanism result values and parameters must be explained and discussed properly.

(b) In Result Section T1, T2, …. Are mentioned what are these? Mention and explain them properly. Also, in Table 1 and Table 3 why only top 5 classes or least 5 classes are mentioned. Description about other classes?

6) The acknowledgment, funding, and data availability sections are included, but their formatting should match journal requirements.

7) Some recent and relevant works could be included to broaden the comparison and highlight novelty.

8) On Page 2, Line 28: the phrase “will make this more problematic” is awkwardly worded. Consider rephrasing as “further exacerbates the problem.”

9) Some sentences are very long; breaking them into shorter sentences would enhance readability.

Please revise the manuscript according to the above points and resubmit. Addressing these issues will significantly improve the clarity, rigor, and presentation quality of the paper, enhancing its suitability for final publication.

**Do you want your identity to be public for this peer review?** For information about this choice, including consent withdrawal, please see our Privacy Policy

Reviewer #1: **Yes: ** Dr.Shaik Karimullah

Reviewer #2: No

---

## [Author Response · Author response to Decision Letter 1]

14 Sep 2025

Thank you for your time and the valuable comments provided during the review process of our manuscript titled ‘Attention to Detail: A Conditional Multi-Head Transformer for Traffic Sign Recognition.’ Below are the reviewers’ comments and our responses.

Reviewer#1, Concern # 1: The sentence structure in abstract need to be improve i-e. “While issues related to object recognition, such as traffic sign recognition… “ is too lengthy and contains overly technical descriptions. Abstract should instead briefly state the problem, the novelty of the approach, key results, and implications.

a. Phrases like “gratuitous” are misused and should be replaced with clearer academic wording.

Author response: Thank you for the valuable comment.

Author action: The abstract has been restructured for conciseness, technical wording refined, and “gratuitous” replaced with clearer academic phrasing.

Reviewer#1, Concern # 2: The introduction is informative but contains repetition (e.g., multiple mentions of misclassification risks and safety-critical concerns). Restructuring into a concise flow would improve readability.

a. Sentence like “…a misplaced label can be immensely disastrous will make this more problematic” is grammatically awkward and should be revised for clarity.

Author response: Thank you for highlighting this. We agree that the introduction required refinement.

Author action: The introduction has been restructured to remove repetitions, improve flow, and the awkward sentence has been revised for grammatical clarity.

Reviewer#1, Concern # 3: The listed contributions are significant; however, they should be rewritten more concisely and distinctly in a separate heading, avoiding overlap with methodology descriptions.

a. Also, explain the third contribution, “Proposed CViT incorporates a specialized attention module… “What is meant by specialized attention module here? And how is it specialized?

Author response: Thank you for this helpful observation.

Author action: Contributions have been presented concisely under a separate heading “2 Research Contributions “, and the specialized attention module has been explained as a dynamic mechanism that adjusts attention weights across traffic sign types.

Reviewer#1, Concern #4: The methodology is presented in detail with equations and figures, which is commendable. However:

a. Several equations (e.g., Eq. (1), Eq. (2)–(4)) lack variable definitions. Every parameter (e.g., P, C, d) should be explicitly explained, as this can confuse readers unfamiliar with the notation.

b. In Eq. (5) bg is not described, what is bg and what is its use in Eq. 5?

c. In 3.1.1 “Then each patch x_i for(i=1,2,…,N) are flattened into a vector of size P^2∙C, where C is the number of channels, for example, for RGB images C=3.” is mentioned but again repeated at the end of 3.1.2. It needs explanation, why is it repeated again?

Author response: Thank you for the detailed suggestions. We acknowledge the need for clarity.

Author action: Variable definitions have been added to all equations, bg has been defined and its role explained in Eq. (5), and the repeated flattening step has been clarified to indicate its connection to positional encoding.

Reviewer#1, Concern # 5: Several grammar issues (e.g., “Figure 6 demonstrate” should be “demonstrates”). The manuscript needs a thorough language polish.

Author response: Thank you for pointing this out.

Author action: The entire manuscript has undergone thorough proofreading and grammar polishing.

Reviewer#1, Concern # 6: Ensure that acronyms (e.g., GAP, FFN, MHA) are introduced at first use and used consistently throughout.

Author response: Thank you for the observation.

Author action: Acronyms have been introduced at first mention and standardized throughout the paper.

Reviewer#1, Concern # 7: Equation numbering is inconsistent, like in Eq. 1 to Eq. 10, parentheses “()” are used, but in Eq. 11-12, square braces “[ ]” are used.

Author response: Thank you for noticing this.

Author action: Equation numbering has been corrected and made consistent across the manuscript.

Reviewer#1, Concern # 8: In Section 3.1.1, the phrase “as e.g., 224×224” is unpolished. Suggested edit: “e.g., resized to 224×224 pixels.”

Author response: Thank you for highlighting this.

Author action: The phrase has been revised to “These images are first resized to a fixed resolution as H*W e.g., 224×224 pixels.

Reviewer#1, Concern # 9: Reference formatting is inconsistent: some include DOIs, others do not. Standardize per journal guidelines.

Author response: Thank you for pointing this out.

Author action: All references have been standardized according to the journal’s guidelines, with DOIs included where available.

Reviewer#2, Concern # 1: 1) (a) The related work section is extensive; however, the flow is occasionally interrupted by abrupt transitions. In addition, it lacks critical analysis and a discussion of the limitations of prior studies. It is also unclear why ViT is specifically preferred or emphasized.

(b) Furthermore, please add a new table in the related work section that summarizes the pros and cons of previous methods and clearly illustrates how your contribution relates to them and provides improvements.

Author response: Thank you for this constructive feedback.

Author action: The related work section has been restructured with smoother transitions, critical analysis, and justification of ViT usage. A new summary table (Table 1) comparing prior methods with our contribution has been included.

Reviewer#2, Concern # 2: In Figure 4, what is *N? is it mistaken? if not, please properly explain its working or importance in Figure 4.

Author response: Thank you for the observation.

Author action: *N has been defined as the total number of patches, and its role in Figure 4 has been explained in the caption and text.

Reviewer#2, Concern # 3: In 3.2.4, “The Residual & Normalization after Attention, Feed-Forward Network (FFN), and Residual & Normalization after FFN are mathematically represented …” are these the modified equations or just the basic ones? If modified, then please explain them. And if these are just the same as the basic Vision transformer, then please include references.

Author response: Thank you for pointing this out.

Author action: We clarified that these equations are standard transformer equations and added appropriate references.

Reviewer#2, Concern #4: The algorithm of the proposed Conditional Vision Transformer is good and well describes the whole methodology. The fail-control mechanism is a key strength, but its explanation is somewhat limited, and it is not discussed in the abstract.

(a) Table 7 includes valuable metrics, but abbreviations (e.g., “Critical Omissions,” “Error Reduction”) should be explained in captions.

(b) Figure 5 needs to be explained properly with details of metrics and parameters.

Author response: Thank you for emphasizing this important point.

Author action: Fail-control explanation has been expanded in the methodology and included in the abstract. Table 7 caption now explains abbreviations, and Figure 5 description has been enhanced with metric and parameter details.

Reviewer#2, Concern # 5: There are typos in the experimental section may be like “color jitter”, “WeightedRandomSampler”. It also misses the details about the training-testing ratio for the experiments.

(a) Fail Control Mechanism result values and parameters must be explained and discussed properly.

(b) In Result Section T1, T2, …. Are mentioned what are these? Mention and explain them properly. Also, in Table 1 and Table 3 why only top 5 classes or least 5 classes are mentioned. Description about other classes?

Author response: Thank you for the detailed review.

Author action: Typos have been corrected, train-test split (80:10:10) added, Fail Control parameters explained, T1–T43 notation clarified as class IDs, and explanation provided for focusing on top/bottom 5 classes while noting full results are available.

Reviewer#2, Concern # 6: The acknowledgment, funding, and data availability sections are included, but their formatting should match journal requirements.

Author response: Thank you for pointing this out.

Author action: These sections have been reformatted to align with journal requirements.

Reviewer#2, Concern # 7: Some recent and relevant works could be included to broaden the comparison and highlight novelty.

Author response: Thank you for this valuable suggestion.

Author action: Five recent works (2024–2025) on lightweight ViTs, adversarial robustness, autoencoder defenses, cross-domain TSR, and multimodal methods have been added to the related work section with critical discussion.

Reviewer#2, Concern # 8: On Page 2, Line 28: the phrase “will make this more problematic” is awkwardly worded. Consider rephrasing as “further exacerbates the problem.”

Author response: Thank you for the suggestion.

Author action: The phrase has been revised.

Reviewer#2, Concern # 9: Some sentences are very long; breaking them into shorter sentences would enhance readability.

Author response: Thank you for pointing this out.

Author action: Long sentences have been revised and broken into shorter, clearer statements to improve readability.

---

## [Decision Letter · Decision Letter 1]

24 Sep 2025

Dear Dr.,

Thank you for submitting your manuscript to PLOS ONE. After careful consideration, we feel that it has merit but does not fully meet PLOS ONE’s publication criteria as it currently stands. Therefore, we invite you to submit a revised version of the manuscript that addresses the points raised during the review process.

We look forward to receiving your revised manuscript.

Kind regards,

Mudassir Khan, Ph.D

Academic Editor

PLOS ONE

Journal Requirements:

Additional Editor Comments (if provided):

Reviwer1 and reviewer2 asked the authors to improve this manuscript significantly. I hope that the authors improve this manuscript following the comments.

Reviewers' comments:

Reviewer's Responses to Questions

**Comments to the Author**

Reviewer #1: All comments have been addressed

Reviewer #2: All comments have been addressed

2. Is the manuscript technically sound, and do the data support the conclusions?

Reviewer #1: Yes

Reviewer #2: Yes

3. Has the statistical analysis been performed appropriately and rigorously?

Reviewer #1: Yes

Reviewer #2: Yes

4. Have the authors made all data underlying the findings in their manuscript fully available?

Reviewer #1: Yes

Reviewer #2: Yes

5. Is the manuscript presented in an intelligible fashion and written in standard English?

Reviewer #1: Yes

Reviewer #2: Yes

Reviewer #1: The manuscript presents a well-structured study with a novel Conditional Vision Transformer (CViT) for traffic sign recognition. The work is technically sound, and experimental results are impressive. Only a few minor revisions are suggested for clarity, formatting, and presentation.

1. The abstract is clear but slightly long. Consider shortening by removing repeated phrases, especially where results are restated in the last two sentences.

2. The diagrams are informative, but the resolution and font size are low. Please improve image quality and enlarge labels so they are easily readable.

3. There are small grammatical issues and inconsistent tenses (e.g., “in life-or-death scenarios like autonomous driving, but in safety-critical scenarios…”). A careful language edit would improve clarity.

4. The comparison with recent models (Figure 6) is strong, but adding a short explanation of why CViT outperforms others (e.g., adaptive attention vs. fixed attention) would provide better insight.

5. Some entries are missing DOIs (e.g., refs. [31], [33], [35]). Please revise to follow PLOS ONE reference formatting guidelines.

6. Kindly read and cite these references:

1. Sun, G., Song, L., Yu, H., Chang, V., Du, X.,... Guizani, M. (2019). V2V Routing in a VANET Based on the Autoregressive Integrated Moving Average Model. IEEE Transactions on Vehicular Technology, 68(1), 908-922. doi: 10.1109/TVT.2018.2884525

2. Sun, G., Zhang, Y., Liao, D., Yu, H., Du, X.,... Guizani, M. (2018). Bus-Trajectory-Based Street-Centric Routing for Message Delivery in Urban Vehicular Ad Hoc Networks. IEEE Transactions on Vehicular Technology, 67(8), 7550-7563. doi: 10.1109/TVT.2018.2828651

3. Li, Z., Hu, J., Leng, B., Xiong, L., & Fu, Z. (2024). An Integrated of Decision Making and Motion Planning Framework for Enhanced Oscillation-Free Capability. IEEE Transactions on Intelligent Transportation Systems, 25(6), 5718-5732. doi: 10.1109/TITS.2023.3332655

4. Shen, Z., He, Y., Du, X., Yu, J., Wang, H.,... Wang, Y. (2024). YCANet: Target Detection for Complex Traffic Scenes Based on Camera-LiDAR Fusion. IEEE Sensors Journal, 24(6), 8379-8389. doi: 10.1109/JSEN.2024.3357826

5. Wang, J., Wang, H., Song, J., Chen, X., Guo, J., Li, K.,... Huang, B. (2025). Knowledge-guided self-learning control strategy for mixed vehicle platoons with delays. Nature Communications, 16(1), 7705. doi: 10.1038/s41467-025-62597-x

6. Lu, Y., Chen, S., Zhang, X., Pan, X., Gang, Y.,... Wang, C. (2025). A quantum-enhanced heuristic algorithm for optimizing aircraft landing problems in low-altitude intelligent transportation systems. Scientific Reports, 15(1), 21606. doi: 10.1038/s41598-025-05261-0

7. Song, D., Zhao, J., Zhu, B., Han, J., & Jia, S. (2024). Subjective Driving Risk Prediction Based on Spatiotemporal Distribution Features of Human Driver’s Cognitive Risk. IEEE Transactions on Intelligent Transportation Systems, 25(11), 16687-16703. doi: 10.1109/TITS.2024.3409874

8. Zuo, C., Zhang, X., Zhao, G., & Yan, L. (2025). PCR: A Parallel Convolution Residual Network for Traffic Flow Prediction. IEEE Transactions on Emerging Topics in Computational Intelligence, 9(4), 3072-3083. doi: 10.1109/TETCI.2025.3525656

9. Sun, T., Guo, R., Chen, G., Wang, H., Li, E.,... Zhang, W. (2025). RID-LIO: robust and accurate intensity-assisted LiDAR-based SLAM for degenerated environments. Measurement Science and Technology, 36(3), 36313. doi: 10.1088/1361-6501/adb769

10. Zhou, X., Zhao, Z., Shen, J., Liu, Z., Liu, Y.,... Xue, B. (2025). Sparse Aperture ISAR Autofocusing and Imaging Algorithm Based on Log-Sum Regularization. IEEE Transactions on Geoscience and Remote Sensing, 63. doi: 10.1109/TGRS.2025.3560140

11. Zhu, B., Tang, R., Zhao, J., Zhang, P., Li, W., Cao, X.,... Li, S. (2025). Critical scenarios adversarial generation method for intelligent vehicles testing based on hierarchical reinforcement architecture. Accident Analysis & Prevention, 215, 108013. doi: https://doi.org/10.1016/j.aap.2025.108013

Reviewer #2: Minor Revision:

The authors have carefully addressed the major revision comments, and the manuscript is now much improved. The paper is well explained, and the experimental results are convincing. However, a few minor corrections are still required to further enhance clarity and consistency in presentation:

1. Please ensure that “Conditional Visual Transformer (CViT)” is written in full only at its first occurrence, and that all subsequent mentions use the abbreviation “CViT” consistently throughout the manuscript (including figures, tables, and supplementary material).

2. Ensure that “Convolutional Neural Networks (CNNs)” and “Vision Transformers (ViTs)” are expanded only at their first occurrence, and that all subsequent mentions use the abbreviations “CNNs” and “ViTs” consistently throughout the manuscript.

3. In the Conclusion section, please add the proposed results explicitly, with separate values for accuracy, precision, recall, and F1-score for easier understanding by readers. Moreover, kindly include 2–3 sentences highlighting possible future research directions based on your work.

4. In the Experimental Setup and Results section, add the total number of parameters used in this study to provide a clearer picture of the model’s complexity and computational requirements.

5. Table 6 is presented in the manuscript, but its caption and detailed information are not adequately discussed in the text. Please revise the manuscript to reference and explain the content of Table 6 clearly within the main body of the paper.

6. Please carefully review the numbering of all equations and their citations throughout the manuscript. Some inconsistencies were noted in equation numbering and flow; these should be corrected for accuracy and clarity.

Once these minor issues are corrected, the manuscript will be suitable for publication.

**Do you want your identity to be public for this peer review?** For information about this choice, including consent withdrawal, please see our Privacy Policy

Reviewer #1: No

Reviewer #2: No

---

## [Author Response · Author response to Decision Letter 2]

29 Sep 2025

REVIEWERS COMMENTS:

We sincerely thank the editors and reviewers for their thoughtful and constructive comments. Their feedback has been very helpful in improving the clarity, rigor, and overall quality of our manuscript. We have carefully revised the paper in line with each point raised, and our detailed responses are provided below:

Reviewer#1, Concern # 1: The abstract is clear but slightly long. Consider shortening by removing repeated phrases, especially where results are restated in the last two sentences.

Author response: Thank you for the valuable comment.

Author action: The abstract has been shortened by removing repeated phrases and redundant result restatements.

Reviewer#1, Concern # 2: The diagrams are informative, but the resolution and font size are low. Please improve image quality and enlarge labels so they are easily readable.

Author response: Thank you for highlighting this.

Author action: All diagrams have been updated with higher resolution and enlarged font sizes for readability.

Reviewer#2, Concern # 3: There are small grammatical issues and inconsistent tenses (e.g., “in life-or-death scenarios like autonomous driving, but in safety-critical scenarios…”). A careful language edit would improve clarity.

Author response: Thank you for this helpful observation.

Author action: The manuscript has been carefully edited for grammar, tense consistency, and clarity.

Reviewer#1, Concern #4: The comparison with recent models (Figure 6) is strong, but adding a short explanation of why CViT outperforms others (e.g., adaptive attention vs. fixed attention) would provide better insight.

Author response: Thank you for the detailed suggestions. We acknowledge the need for clarity.

Author action: A short explanation highlighting CViT’s adaptive attention mechanism versus fixed attention has been added with the explanation of Figure 6.

Reviewer#1, Concern # 5: Some entries are missing DOIs (e.g., refs. [31], [33], [35]). Please revise to follow PLOS ONE reference formatting guidelines.

Author response: Thank you for pointing this out.

Author action: All missing DOIs have been added, and references reformatted according to PLOS ONE guidelines.

Reviewer#1, Concern # 6: . Kindly read and cite these references:

1. Sun, G., Song, L., Yu, H., Chang, V., Du, X.,... Guizani, M. (2019). V2V Routing in a VANET Based on the Autoregressive Integrated Moving Average Model. IEEE Transactions on Vehicular Technology, 68(1), 908-922. doi: 10.1109/TVT.2018.2884525

2. Sun, G., Zhang, Y., Liao, D., Yu, H., Du, X.,... Guizani, M. (2018). Bus-Trajectory-Based Street-Centric Routing for Message Delivery in Urban Vehicular Ad Hoc Networks. IEEE Transactions on Vehicular Technology, 67(8), 7550-7563. doi: 10.1109/TVT.2018.2828651

3. Li, Z., Hu, J., Leng, B., Xiong, L., & Fu, Z. (2024). An Integrated of Decision Making and Motion Planning Framework for Enhanced Oscillation-Free Capability. IEEE Transactions on Intelligent Transportation Systems, 25(6), 5718-5732. doi: 10.1109/TITS.2023.3332655

4. Shen, Z., He, Y., Du, X., Yu, J., Wang, H.,... Wang, Y. (2024). YCANet: Target Detection for Complex Traffic Scenes Based on Camera-LiDAR Fusion. IEEE Sensors Journal, 24(6), 8379-8389. doi: 10.1109/JSEN.2024.3357826

5. Wang, J., Wang, H., Song, J., Chen, X., Guo, J., Li, K.,... Huang, B. (2025). Knowledge-guided self-learning control strategy for mixed vehicle platoons with delays. Nature Communications, 16(1), 7705. doi: 10.1038/s41467-025-62597-x

6. Lu, Y., Chen, S., Zhang, X., Pan, X., Gang, Y.,... Wang, C. (2025). A quantum-enhanced heuristic algorithm for optimizing aircraft landing problems in low-altitude intelligent transportation systems. Scientific Reports, 15(1), 21606. doi: 10.1038/s41598-025-05261-0

7. Song, D., Zhao, J., Zhu, B., Han, J., & Jia, S. (2024). Subjective Driving Risk Prediction Based on Spatiotemporal Distribution Features of Human Driver’s Cognitive Risk. IEEE Transactions on Intelligent Transportation Systems, 25(11), 16687-16703. doi: 10.1109/TITS.2024.3409874

8. Zuo, C., Zhang, X., Zhao, G., & Yan, L. (2025). PCR: A Parallel Convolution Residual Network for Traffic Flow Prediction. IEEE Transactions on Emerging Topics in Computational Intelligence, 9(4), 3072-3083. doi: 10.1109/TETCI.2025.3525656

9. Sun, T., Guo, R., Chen, G., Wang, H., Li, E.,... Zhang, W. (2025). RID-LIO: robust and accurate intensity-assisted LiDAR-based SLAM for degenerated environments. Measurement Science and Technology, 36(3), 36313. doi: 10.1088/1361-6501/adb769

10. Zhou, X., Zhao, Z., Shen, J., Liu, Z., Liu, Y.,... Xue, B. (2025). Sparse Aperture ISAR Autofocusing and Imaging Algorithm Based on Log-Sum Regularization. IEEE Transactions on Geoscience and Remote Sensing, 63. doi: 10.1109/TGRS.2025.3560140

11. Zhu, B., Tang, R., Zhao, J., Zhang, P., Li, W., Cao, X.,... Li, S. (2025). Critical scenarios adversarial generation method for intelligent vehicles testing based on hierarchical reinforcement architecture. Accident Analysis & Prevention, 215, 108013. doi: https://doi.org/10.1016/j.aap.2025.108013

Author response: Thank you for the suggestions.

Author action: The most relevant suggested references ([39-43]) have been cited at appropriate points in the manuscript.

Reviewer#2, Concern # 1: Please ensure that “Conditional Visual Transformer (CViT)” is written in full only at its first occurrence, and that all subsequent mentions use the abbreviation “CViT” consistently throughout the manuscript (including figures, tables, and supplementary material).

Author response: Thank you for the valuable comment.

Author action: CViT has been written in full only at the first occurrence and abbreviated consistently throughout the manuscript.

Reviewer#2, Concern # 2: Ensure that “Convolutional Neural Networks (CNNs)” and “Vision Transformers (ViTs)” are expanded only at their first occurrence, and that all subsequent mentions use the abbreviations “CNNs” and “ViTs” consistently throughout the manuscript.

Author response: Thank you for highlighting this.

Author action: CNNs and ViTs are expanded only at their first occurrence; abbreviations are used consistently throughout.

Reviewer#2, Concern # 3: In the Conclusion section, please add the proposed results explicitly, with separate values for accuracy, precision, recall, and F1-score for easier understanding by readers. Moreover, kindly include 2–3 sentences highlighting possible future research directions based on your work.

Author response: Thank you for this helpful observation.

Author action: Accuracy, precision, recall, and F1-score are now explicitly presented in the Conclusion, along with future research directions.

Reviewer#2, Concern #4: In the Experimental Setup and Results section, add the total number of parameters used in this study to provide a clearer picture of the model’s complexity and computational requirements.

Author response: Thank you for the detailed suggestions. We acknowledge the need for clarity.

Author action: The total parameter count of the model has been added in the Experimental Setup and Results section.

Reviewer#2, Concern # 5: Table 6 is presented in the manuscript, but its caption and detailed information are not adequately discussed in the text. Please revise the manuscript to reference and explain the content of Table 6 clearly within the main body of the paper.

Author response: Thank you for pointing this out.

Author action: The text has been revised to clearly reference and explain the content of Table 6.

Reviewer#2, Concern # 6: Please carefully review the numbering of all equations and their citations throughout the manuscript. Some inconsistencies were noted in equation numbering and flow; these should be corrected for accuracy and clarity.

Author response: Thank you for the observation.

Author action: Equation numbering and citations have been carefully reviewed and corrected for accuracy and flow.

---

## [Decision Letter · Decision Letter 2]

9 Oct 2025

Attention to Detail: A Conditional Multi-Head Transformer for Traffic Sign Recognition

PONE-D-25-39099R2

Dear Author,

We’re pleased to inform you that your manuscript has been judged scientifically suitable for publication and will be formally accepted for publication once it meets all outstanding technical requirements.

Kind regards,

Mudassir Khan, Ph.D

Academic Editor

PLOS ONE

Additional Editor Comments (optional):

Dear Editor,

Thank you for the opportunity to review the manuscript titled “Attention to Detail: A Conditional Multi-Head Transformer for Traffic Sign Recognition.

Overall, I found the manuscript to be well-written and informative.

Overall, I believe that this manuscript has the potential to make a valuable contribution to the field. However, I recommend that the authors address all the concerns raised during the review process. My decision is to accept in the present form.

Regards,

Dr Mudassir Khan

Reviewers' comments:

Reviewer's Responses to Questions

**Comments to the Author**

Reviewer #1: All comments have been addressed

Reviewer #2: All comments have been addressed

2. Is the manuscript technically sound, and do the data support the conclusions?

Reviewer #1: Yes

Reviewer #2: Yes

3. Has the statistical analysis been performed appropriately and rigorously?

Reviewer #1: Yes

Reviewer #2: Yes

4. Have the authors made all data underlying the findings in their manuscript fully available?

Reviewer #1: Yes

Reviewer #2: Yes

5. Is the manuscript presented in an intelligible fashion and written in standard English?

Reviewer #1: Yes

Reviewer #2: Yes

Reviewer #1: All the Comments were pefectly addressed in manuscript and the proposed methodology gives possible solutions in the research field.

Reviewer #2: The authors addressed every comment. They have thoroughly and satisfactorily addressed every point that I raised. The manuscript is acceptable in its current form.

**Do you want your identity to be public for this peer review?** For information about this choice, including consent withdrawal, please see our Privacy Policy

Reviewer #1: No

Reviewer #2: No

---

## [Editor Report · Acceptance letter]

PONE-D-25-39099R2

PLOS ONE

Dear Dr. Saeed,

I'm pleased to inform you that your manuscript has been deemed suitable for publication in PLOS ONE. Congratulations! Your manuscript is now being handed over to our production team.

Kind regards,

on behalf of

Dr. Mudassir Khan

Academic Editor

PLOS ONE